# Food Protein Sterylation: Chemical Reactions between Reactive Amino Acids and Sterol Oxidation Products under Food Processing Conditions

**DOI:** 10.3390/foods9121882

**Published:** 2020-12-17

**Authors:** Franks Kamgang Nzekoue, Thomas Henle, Giovanni Caprioli, Gianni Sagratini, Michael Hellwig

**Affiliations:** 1School of Pharmacy, University of Camerino, Via Sant Agostino 1, 62032 Camerino, Italy; astride.kamgang@unicam.it (F.K.N.); giovanni.caprioli@unicam.it (G.C.); 2Chair of Food Chemistry, Technische Universität Dresden, 01062 Dresden, Germany; Thomas.Henle@tu-dresden.de (T.H.); Michael.Hellwig@tu-dresden.de (M.H.); 3Institute of Food Chemistry, Technische Universität Braunschweig, Schleinitzstraße 20, 38106 Braunschweig, Germany

**Keywords:** cholesterol, phytosterols, oxysterols, oxyphytosterols, lysine, amino acids, lipation, sterylation, protein modification

## Abstract

Sterols, especially cholesterol and phytosterols, are important components of food lipids. During food processing, such as heating, sterols, like unsaturated fatty acids, can be oxidized. Protein modification by secondary products of lipid peroxidation has recently been demonstrated in food through a process called lipation. Similarly, this study was performed to assess, for the first time, the possibility of reactions between food proteins and sterol oxidation products in conditions relevant for food processing. Therefore, reaction models consisting of oxysterol (cholesterol 5α,6α-epoxide) and reactive amino acids (arginine, lysine, and methionine) were incubated in various conditions of concentration (0–8 mM), time (0–120 min), and temperature (30–180 °C). The identification of lysine adducts through thin-layer chromatography (TLC), high-performance liquid chromatography (HPLC) with a diode array detector (DAD), and electrospray ionization (ESI) mass spectrometry (MS) evidenced a reaction with lysine. Moreover, the HPLC-ESI with tandem mass spectrometry (MS/MS) analyses allowed observation of the compound, whose mass to charge ratio *m/z* 710.5 and fragmentation patterns corresponded to the reaction product [M + H]^+^ between cholesterol-5α,6α-epoxide and the ε-amino-group of N_α_-benzoylglycyl-l-lysine. Moreover, kinetic studies between N_α_-benzoylglycyl-l-lysine as a model for protein-bound lysine and cholesterol 5α,6α-epoxide were performed, showing that the formation of lysine adducts strongly increases with time, temperature, and oxysterol level. This preliminary study suggests that in conditions commonly reached during food processing, sterol oxidation products could react covalently with protein-bound lysine, causing protein modifications.

## 1. Introduction

Cholesterol is an important constituent of animal cells, and is a precursor for the synthesis of steroid hormones, bile acids, and vitamin D [1]. The main dietary sources of cholesterol include eggs, meat (poultry, pork, and beef), butter, cheese, and full-fat dairy products [2].

Phytosterols (PSs) are bioactive compounds that are structurally similar to cholesterol and commonly found in plants [3]. Because of their possible health benefits [4], PSs are expansively requested and exploited as bioactive ingredients to produce functional foods [5]. PS content is high in nuts, legumes, and cereals (0.035 to 0.12 g/100 g). However, the highest levels are found in unrefined vegetable oils, such as corn oil (1.1 g/100 g), and PS-enriched foods, such as functional margarine (10–12 g/100 g) [6].

Like unsaturated lipids, cholesterol and PSs are prone to oxidation in food, leading to the formation of a class of compounds named cholesterol oxidation products (COPs) and PS oxidation products (POPs), respectively [7]. These compounds demonstrate cytotoxic, pro-inflammatory, and pro-atherogenic effects [8]. The main factors influencing the oxidation of cholesterol and PSs include temperature, time, sterol concentration, and food composition [9,10]. As an example, PS-enriched margarine heated to 180 °C for 0–10 min showed an increase of POP content from 23 to 1874 µg/g [11].

The conditions promoting the formation of COP/POP, also stimulate chemical reactions in food, such as the Maillard reaction, which is a non-enzymatic series of reactions between reducing sugars and free amino groups of proteins [12]. Although is it considered the most important reaction in food, the Maillard reaction is not the only reaction responsible for protein modification. Indeed, the nucleophilic addition of amino acids, such as lysine, to lipid peroxidation products, such as reactive aldehydes (RAs), is another important mechanism of protein modification [13]. This series of complex reactions named “lipation” [14] was recently discovered in food and became a new topic of research in food chemistry. As an example, lipation products from the reaction between 4-hydroxy-2-nonenal (4-HNE) and the amino group of lysine were observed in peanuts during roasting [15].

Typical mechanisms of protein modifications in foods involve the nucleophilic attack of free ε-amino groups in proteins with reactive functional groups in other food components, such as aldehydes in glucose or oxidized fatty acids [12,15].

Protein modifications induced by reactions such as the Maillard reaction are responsible for the allergies observed for many foods after processing [16]. Indeed, various studies reported that the glycation of food proteins during food processing enhanced their potential allergenicity and immunogenicity [17]. In addition, reactions of food proteins with other food components like lipids could explain the common food allergies observed from the processing of tree nuts (e.g., almonds, walnuts, pecans, cashews, hazelnuts, etc.), peanuts, or soybeans [15,18]. Due to the extent of food allergy prevalence in the world, it is important to understand all the structural modifications of food proteins by assessing possible interactions with components present in the food matrix. Moreover, studies on the kinetics of protein modifications could help to understand the conditions promoting or reducing the allergenicity of food proteins.

Food processing, such as roasting, frying, and baking [8,19,20], commonly used with foods rich in cholesterol and PS, can speed up the production of high levels of COPs/POPs, including 7α/β-hydroxysterols, 7-ketosterols, 5α,6α-/5β,6β-epoxysterols, and 3β,5α,6β-triols (Figure 1). Due to the reactivity of carbonyl and epoxy derivatives, we hypothesized that COPs and POPs could react like RAs with the nucleophilic amino acid side chains of food proteins. Indeed, protein modifications induced by COPs and POPs could play a role in food allergies observed in foods with high sterol content, such as egg yolks, nuts, and soybeans. However, regarding reactions in food, no information is available concerning possible protein interaction with COPs/POPs.

Therefore, this preliminary study aims to evaluate, for the first time, the possibility for reactions between amino acid side chains and POPs/COPs under conditions relevant to food processing. Due to the lack of commercially available POP standards, a COP standard was used in this study. Model mixtures consisting of cholesterol-5α,6α-epoxide and relevant reactive amino acids were thus studied to assess the formation of oxysterol–amino acid adducts. Potential reactions were investigated through thin-layer chromatography (TLC) and high-performance liquid chromatography (HPLC). Mass spectrometry (MS) is a suitable analytical technique for revealing the formation of new compounds during chemical reactions and helping in the elucidation of unknown compounds’ structures. Therefore, the supposed reaction products were confirmed and elucidated by HPLC, electrospray ionization (ESI), and tandem mass spectrometry (MS/MS). Furthermore, by performing kinetic studies, the reaction behavior between cholesterol 5α,6α-epoxide and lysine was studied through gas chromatography (GC) and HPLC analyses.

## 2. Materials and Methods

### 2.1. Materials

β-Sitosterol (C_29_H_50_O, CAS No. 83-46-5), cholesterol (C_27_H_46_O, CAS No. 57-88-5), and cholesterol 5α,6α-epoxide (C_27_H_46_O_2_, CAS No. 1250-95-9) were supplied by Merck (Milan, Italy). N_α_-Boc-l-lysine, N_α_-benzoylglycyl-l-lysine, N_α_-Boc-l-methionine, and N_α_-Boc-l-arginine were obtained from Bachem (Bubendorf, Switzerland). Methanol, ethyl acetate, hexane, chloroform, and trifluoroacetic acid were obtained from Merck (Darmstadt, Germany). Acetonitrile and Tri-Sil (HDMS:TMCS:Pyridine) reagents were obtained from Fischer Scientific (Schwerte, Germany). All chemicals were of the highest purity. For all experiments, double-distilled water was used (Destamat Bi 18E; QCS GmbH, Maintal, Germany).

### 2.2. Incubations of Amino Acids with Sterols/Oxysterols

To assess the possibility of amino acid–COP/POP reactions, model mixtures made of protected reactive amino acids (Boc-arginine, Boc-lysine, and Boc-methionine) and oxysterol (cholesterol 5α,6α-epoxide) at various concentrations (0–8 mM) were incubated in different conditions (temperatures: 30–180 °C, time: 0–120 min). Standard solutions of amino acids were prepared in methanol, while oxysterol was prepared in ethyl acetate.

### 2.3. Kinetic Studies of the Reaction between Cholesterol 5α,6α-Epoxide and Nα-Benzoylglycyl-l-Lysine

N_α_-benzoylglycyl-l-lysine and cholesterol 5α,6α-epoxide were mixed and incubated in various conditions to study the effects of time, temperature, and concentrations on amino acid–COP/POP reactions.

#### 2.3.1. Effect of Time

Samples of 4 mM N_α_-benzoylglycyl-l-lysine, 4 mM cholesterol 5α,6α-epoxide, and their mixture (4 mM in methanol/ethyl acetate, 50:50) were incubated at 180 °C. Before incubation (T0) and after 5, 10, 30, 60, and 120 min, an aliquot of 100 μL was collected for analysis.

#### 2.3.2. Effect of Temperature

The samples were incubated at different temperatures: 30, 60, 90, 120, 150, and 180 °C. After 120 min, incubations were stopped and samples were collected for analysis.

#### 2.3.3. Effect of Concentrations

A fixed concentration of N_α_-benzoylglycyl-l-lysine (4 mM) was mixed with various concentrations of cholesterol 5α,6α-epoxide (0, 0.4, 0.8, 2, 4, and 8 mM). Samples with various concentration proportions (epoxide/lysine: 0:1, 0.1:1, 0.2:1, 0.5:1, 1:1, and 2:1) were thus obtained and incubated at 180 °C for 120 min.

#### 2.3.4. Analyses of Samples

For the monitoring of N_α_-benzoylglycyl-l-lysine and the obtained derivatives, samples were diluted with methanol (1:10 dilution) and filtrated using 0.45 μm filters for HPLC analyses. The monitoring of cholesterol 5α,6α-epoxide was performed through GC flame ionization detector (FID) analyses.

### 2.4. GC-FID Analyses of Sterols and Cholesterol 5α,6α-Epoxide

The analysis of sterols and cholesterol 5α,6α-epoxide was performed following a referenced method [21] with slight modifications. Briefly, samples were dried under nitrogen and reconstituted with 100 µL of Tri-Sil reagent. After silylation (30 min at 80 °C), the derivatized samples were diluted with 400 µL of hexane, filtrated, and analyzed by GC-FID.

GC-FID analysis was performed using a GC 7820A-FID device with an autoinjector (all from Agilent, Böblingen, Germany). The separation of analytes was performed using a Zebron ZB-5 capillary column (30 m × 0.25 mm i.d. × 0.25 µm film thickness, Phenomenex, Aschaffenburg, Germany). Nitrogen was used as a carrier gas with a constant flow of 1 mL/min. The injector temperature was set at 300 °C, and 1 μL of the sample was injected in split mode (split ratio 5:1) with a split flow of 5 mL/min. For GC, the oven temperature was first held at 120 °C and was programmed to increase until 260 °C at a rate of 20 °C/min with an initial isothermal delay of 0.5 min. Then, the temperature was raised until 300 °C at a rate of 2 °C/min. The detector temperature was set at 300 °C.

### 2.5. HPLC Diode Array Detector (DAD) Analysis

For the analysis of amino acids, the monitoring, and the quantification of reaction products, HPLC-DAD analyses were performed. The column used was an Eurospher 100-5 C18 column (250 × 4.6 mm, 5 µm; Knauer, Berlin, Germany) set at 30 °C. The HPLC system (Knauer, Berlin, Germany) consisted of a degasser, a pump, a thermostat, an autosampler, and a diode array detector (DAD). The injection volume was 15 µL at a flow rate of 1 mL/min. Gradient elution was used with solvents A (water/acetonitrile, 90:10) and B (acetonitrile/water, 90/10), both containing 10 mM of trifluoroacetic acid. The solvent composition varied as follows: 0 min, 100% A; 0–5 min, 100% A; 5–15 min 25% A; 15–17 min, 100% A; 17–25 min, 100% A. The detection wavelength was fixed at 254 nm.

### 2.6. HPLC-MS/MS Analysis

The identification of major reaction products between oxysterols and amino acids was performed by HPLC-ESI-MS analyses [22]. The HPLC-ESI-MS system consisted of a binary pump, an autosampler, an online degasser, a column thermostat, and a triple-quadrupole mass spectrometer (G6410 A; all from Agilent Technologies, Böblingen, Germany). The column Zorbax 100 SB-C18 (50 × 2.1 mm, 3.5 μm; Agilent) set at 35 °C was used for the separation of compounds. The mobile phase consisted of water (solvent A) and acetonitrile (solvent B), both containing 10 mM of nonafluoropentanoic acid (NFPA). The solvents were pumped at a flow rate of 0.25 mL/min in a gradient mode (0 min, 95% A; 18 min, 15% A; 22 min, 15% A; 24 min, 95% A; 30 min, 95% A). The injection volume was 5 μL.

The full scanning regime (MS2-Scan) in the range of mass to charge ratio (*m/z)* 100–1000 was used. Analyses were performed in positive ionization mode. The ESI source conditions were as follows: Nitrogen was used as the nebulizing gas (gas flow, 11 L/min; gas temperature, 350 °C; nebulizer pressure, 35 psi), and the capillary voltage was 4000 V. Moreover, product ion scanning was also performed with the conditions given below.

### 2.7. Statistical Analysis

The results are expressed as mean values ± standard deviations (S.D.) of three replicated measurements.

## 3. Results

### 3.1. Identification of Reaction Products between Cholesterol 5α,6α-Epoxide and Amino Acid Side Chains

To study possible reactions between COPs/POPs and amino acid side chains, commercial cholesterol 5α,6α-epoxide standard (as a model for COPs/POPs) was incubated with relevant reactive amino acids, namely arginine, lysine, and methionine. In order to exclude reactions at the amino group, which is not available for reaction in proteins, N-protected amino acids were used. Therefore, N_α_-benzoylglycyl-l-lysine, N_α_-Boc-l-arginine, N_α_-Boc-l-lysine, and N_α_-Boc-l-methionine were mixed with cholesterol 5α,6α-epoxide. The four equimolar reaction mixtures (4 mM) were incubated and analyzed. The obtained results were compared with those from single amino acids (blank samples) incubated in the same conditions without oxysterol.

#### 3.1.1. TLC Analyses

The first studies were performed using TLC to separate potential reaction products from amino acids. After 60 min of incubation at 80 °C, no formation of “new” spots in the TLC was observed. However, by increasing the incubation temperature to 160 °C, spot separations were observed in mixture models containing cholesterol 5α,6α-epoxide with N_α_-benzoylglycyl-l-lysine and N_α_-Boc-l-lysine. Therefore, HPLC analyses were performed to confirm the formation of reaction products (Appendix A).

#### 3.1.2. HPLC-DAD Analyses

Following the TLC results, the equimolar reaction mixture containing N_α_-benzoylglycyl-l-lysine and cholesterol-5α,6α-epoxide was analyzed using HPLC-DAD. The chromophore present in the structure of N_α_-benzoylglycyl-l-lysine allowed specific detection of the starting material and potential reaction products at 254 nm. Figure 2 shows the overlaid chromatograms of the reaction mixture sample and blank sample (N_α_-benzoylglycyl-l-lysine) incubated in the same conditions (60 min, 160 °C). From the chromatogram of the blank sample, we noted the degradation of N_α_-benzoylglycyl-l-lysine with the formation of some derivatives (retention time R.T.: 11.3 min and 13.4 min). However, from the chromatogram of the mixture sample, the degradation of lysine was far more pronounced. This evidenced that degradation was associated with the formation of a new compound (R.T.: 11.7 min) that was not present in the blank sample chromatogram.

#### 3.1.3. HPLC-ESI-MS/MS Analyses

From the obtained results, a covalent reaction between cholesterol-5α,6α-epoxide and the ε-amino-group of lysine was suggested. Therefore, the hypothesized reaction products were sought by HPLC-ESI-MS analyses (Figure 3). Working in positive ionization mode, protonated molecular ions [M + H]^+^ were researched at *m/z* 649.5 and *m/z* 710.5, respectively, which correspond to the suggested adducts from the reactions of cholesterol-5α,6α-epoxide with N_α_-Boc-L-lysine and N_α_-benzoylglycyl-l-lysine, respectively.

Figure 4 shows the extracted ion chromatograms (E.I.C.) of the reaction products and their respective MS spectra, in which molecular ions with *m/z* corresponding to [M + H]^+^ and [M + Na]^+^ were observed.

When performing HPLC-MS/MS analyses on blank samples, the investigated *m/z* was not observed. However, from the analysis of the reaction mixture samples, the sought compounds were detected (Figure 5A). Moreover, the full scan product ion (Figure 5B) of the reaction product between cholesterol-5α,6α-epoxide and N_α_-benzoylglycyl-l-lysine (*m/z* 710.5 = [M + H]^+^) allowed the observation of fragments corresponding to lysine (*m/z* 147.1 = [M + H]^+^) and N_α_-benzoylglycyl-l-lysine (*m/z* 308.2 = [M + H]^+^). Therefore, these results evidenced the epoxycholesterol-lysine reaction and motivated a deepened study on the kinetics of this reaction.

### 3.2. Kinetic Studies of the Reaction between Cholesterol 5α,6α-Epoxide and N_α_-Benzoylglycyl-l-Lysine

To get an insight into the reaction behavior of COPs/POPs with the ε-amino-group of lysine, cholesterol 5α,6α-epoxide was used as a model of COPs/POPs, while N_α_-benzoylglycyl-l-lysine was used as a model for protein-bound lysine. In this reaction mixture model, incubations were performed in various conditions to study the effects of time, temperature, and concentration. After incubations, the effects of these parameters were monitored through HPLC and GC analyses.

#### 3.2.1. Validation of the Analytical Methods

The analytical methods were validated by assessing their linearity, reproducibility, and sensitivity (Table 1).

##### GC-FID Method Validation

The method linearity was determined by analyzing seven concentrations of cholesterol 5α,6α-epoxide (0.025, 0.05, 0.125, 0.25, 0.50, 1.0, and 1.25 mM). The calibration curve obtained showed a coefficient of determination (R^2^) of 0.999, thus indicating the good linearity of the method. The reproducibility was assessed through the relative standard deviation (%RSD) between consecutive analyses (*n* = 5) performed the same day (intra-day reproducibility) and over five consecutive days (inter-day reproducibility). The intra-day reproducibility was 0.6%, while the inter-day reproducibility was 0.8%, thus demonstrating the good reproducibility of the GC-FID method. The method sensitivity was determined by assessing the limit of detection (LOD) and the limit of quantification (LOQ). Signal-to-noise ratios (S/N) of 10:1 and 3:1 were used to estimate the LOD and the LOQ, respectively. The validated method showed the LOD and LOQ of 4.5 μM and 15 μM, respectively.

##### HPLC-DAD Method Validation

The method linearity was determined by analyzing seven concentrations of N_α_-benzoylglycyl-l-lysine (0.05, 0.1, 0.2, 0.4, 0.6, 0.8, and 1.0 mM). The linearity of the method was confirmed from the obtained calibration curve, which showed a coefficient of correlation (R^2^) of 0.995. The method showed an intra-day and inter-day reproducibility of 1.9% and 3.9%, respectively. The LOD was 0.3 μM, while the LOQ was 1.1 μM.

#### 3.2.2. Effect of Time

Figure 6 and Figure 7 show the overlaid chromatograms of N_α_-benzoylglycyl-l-lysine at different times of incubation in blank samples and the reaction mixture, respectively. The amount of cholesterol 5α,6α-epoxide in blank samples was stable over the incubation time, with levels from 4.00 ± 0.03 mM to 3.83 ± 0.02 mM after 120 min of incubation (Appendix A). Contrariwise, N_α_-benzoylglycyl-l-lysine levels decreased (Figure 8a) in blank samples and the reaction mixture. However, the decrease in the reaction mixture was faster than in the blank sample. This can be explained by the reaction of cholesterol 5α,6α-epoxide with the ε-amino-group of lysine (Figure 3). Indeed, Figure 7 showed that the levels of the reaction product increased with the incubation time. The quantification of the reaction product was performed using the calibration curve of N_α_-benzoylglycyl-l-lysine (Figure 9a).

#### 3.2.3. Effect of Temperature

Figure 8b shows the levels of N_α_-benzoylglycyl-l-lysine at increasing temperatures in blank and reaction mixture samples. The amounts of N_α_-benzoylglycyl-l-lysine decreased with increasing temperatures. However, in the reaction mixture, the decrease of N_α_-benzoylglycyl-l-lysine levels was more pronounced at all the incubation temperatures tested. Inversely, the levels of the reaction product increased with temperature increment (Figure 9b).

#### 3.2.4. Effect of Concentration

Figure 9c shows the evolution of the reaction product in different reaction mixture proportions (epoxide/lysine: 0, 0.1, 0.2, 0.5, 1, and 2). These results showed that the reaction between the oxysterol and lysine occurred and proportionally increased with cholesterol 5α,6α-epoxide concentration.

## 4. Discussion

Reactions of COPs/POPs with amino acid side chains were studied using a reaction model consisting of cholesterol 5α,6α-epoxide and N_α_-benzoylglycyl-l-lysine. This study was motivated by the results reported by Globisch et al. [23], in which complex reactions involving amino acids and carbonyl compounds from lipid peroxidation, such as acrolein, were observed. The reactions leading to covalent attachment of products of lipid peroxidation to proteins were named “lipation” and, like reactive carbonyl compounds, COPs/POPs were supposed to form covalent bonds with proteins during food processing [24].

The oxidation of cholesterol and phytosterols in foods produces a large number of oxidation products (hydroxysterols, ketosterols, epoxysterols, and triols) according to the starting sterol species (Figure 1). To assess the possibility for protein modification induced by the interaction with sterol oxidation products (COPs/POPs), specific analytes were selected. Protein modification involves the chemical reaction between nucleophilic amino acids in proteins and electrophile functional groups in other food components. Therefore, models of reactive protein-bound amino acids were incubated with a model of sterol oxidation products.

As models of reactive protein-bound amino acids, protected amino acids with thiol groups (Boc-methionine) and amino groups (Boc-arginine, Boc-lysine, and N_α_-benzoylglycyl-l-lysine) in their side chains were selected. For sterol oxidation products, COPs were considered due to the absence of commercially available POP standards. Cholesterol 5α,6α-epoxide was selected as a model of COPs because carbons in the epoxide group are highly reactive electrophiles.

From the incubated model mixtures of amino acids and cholesterol 5α,6α-epoxide, chemical reactions were only observed in the mixtures with protected lysine (Figure 3). Analyses revealed that sterol oxidation products—especially epoxysterols—could react with proteins under food processing conditions. HPLC-ESI-MS/MS analyses allowed the identification of the main reaction products from the incubation of epoxysterols and lysine (Figure 4 and Figure 5). Moreover, other products were observed from HPLC analyses (Figure 2) from 12 to 13 min, which could correspond to advanced sterylation products. Further analyses will be performed for a deepened characterization of all the reaction products and intermediates.

Under physiological conditions, oxysterol-binding proteins (OSBPs) have been identified in eukaryotic species, from yeasts to humans [25]. Various studies reported that certain OSBPs can be involved in different types of human tumors [26]. Moreover, Zhang et al. [27] described the covalent conjugation of protein and DNA with sterols serving as molecular linkers using Hedgehog autoprocessing. More recently, the non-covalent conjugations of bovine serum albumin with β-sitosterol and stigmasterol were reported to compete against glycation [28].

However, in food or under conditions relevant for food processing, no study has assessed the possibility of a chemical reaction between COPs/POPs and proteins. In the present study, we observed that cholesterol 5α,6α-epoxide can react with the free ε-amino-group of N_α_-benzoylglycyl-l-lysine, leading to the sterylation of lysine. The reaction of the epoxide with lysine should be possible at both C atoms, leading to the formation of two different structures (Figure 3). This could explain the double peaks in the LC-MS chromatograms (Figure 4).

Kinetic studies revealed that this non-enzymatic reaction is influenced by time, temperature, and oxysterol concentration. The reactions observed in these conditions could also happen in food between POPs and proteins due to the structural similarity of POPs with COPs [29]. For example, foods with high PS content (nuts) or PS-enriched foods, such as margarine (10–12 g/100 g) and cheeses (3–5 g/100g), can produce high levels of POPs during cooking, which would covalently interact with food proteins. The results of kinetic studies suggest that in foods, the formation of protein adducts formed from sterol epoxide may occur with higher temperatures and processing times. Therefore, COP/POP–amino acid reaction could be a source of protein modification, which can influence the allergenic potential of foods [14].

## 5. Conclusions

This exploratory study aimed to evaluate the possibility of reactions between amino acids and COPs/POPs under conditions relevant to food processing. The sterylation of the ε-amino-group of lysine was confirmed by HPLC-ESI-MS/MS analyses. As it is analogous to lysine sterylation, the reaction between POPs and amino acids could be called “phytosterylation”. Further studies will be performed to directly identify and quantify sterylation and phytosterylation products in food samples during food processing through HPLC-MS/MS and Quadrupole Time of Flight Q-TOF MS analyses. Moreover, the biological effects of protein modification by COPs/POPs should be also assessed.

## Figures and Tables

**Figure 1 foods-09-01882-f001:**
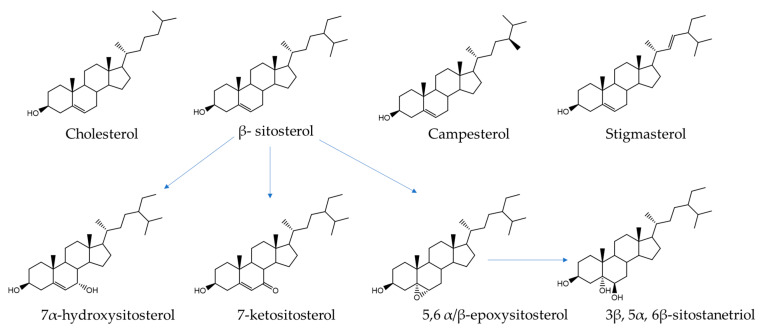
Chemical structures of cholesterol, principal plant sterols, and oxysterols.

**Figure 2 foods-09-01882-f002:**
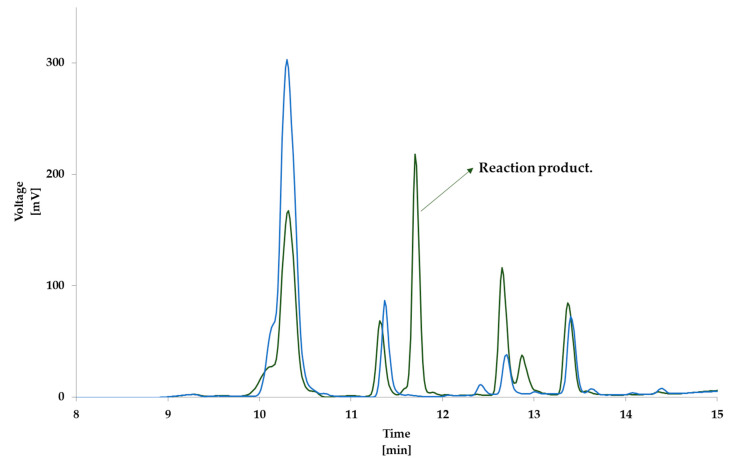
Overlaid chromatograms of N_α_-benzoylglycyl-l-lysine and derivatives in the blank sample (blue) and the reaction mixture (green) incubated in the same conditions (60 min, 160 °C). In the mixture, there is the formation of a reaction product (retention time R.T.: 11.7 min).

**Figure 3 foods-09-01882-f003:**
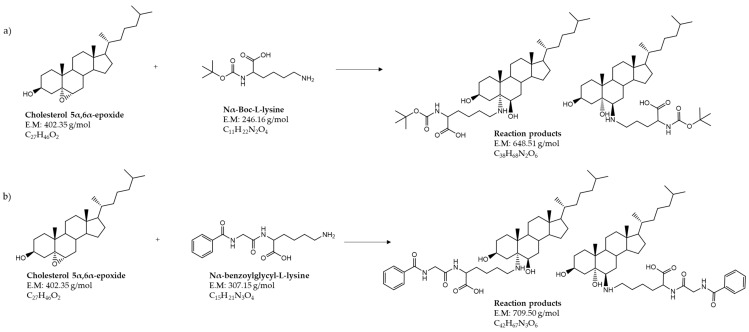
Proposed chemical reactions: (**a**) cholesterol 5α,6α-epoxide and N_α_-Boc-L-lysine; (**b**) cholesterol 5α,6α-epoxide and N_α_-benzoylglycyl-l-lysine. E.M: exact mass (g/mol).

**Figure 4 foods-09-01882-f004:**
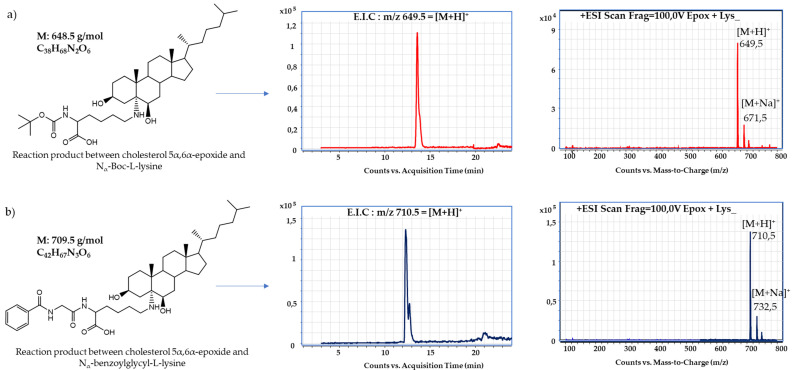
Molecular ions and respective mass spectra of products from reaction mixtures between (**a**) cholesterol 5α,6α-epoxide and N_α_-Boc-l-lysine; (**b**) cholesterol 5α,6α-epoxide and N_α_-benzoylglycyl-l-lysine. E.I.C: Extract ion chromatogram. ESI: Electrospray ionization.

**Figure 5 foods-09-01882-f005:**
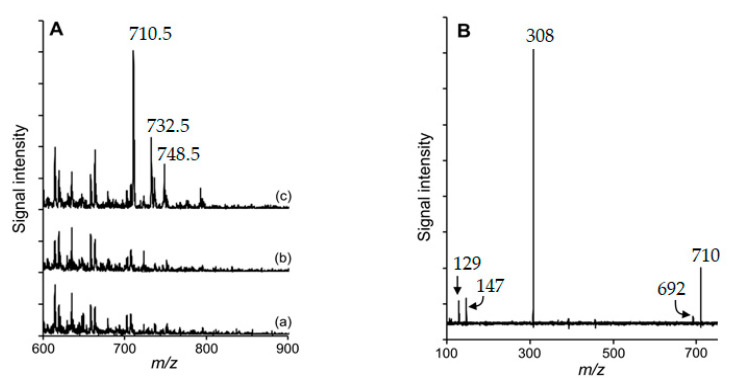
Mass spectrometry (MS) spectra recorded during high-performance liquid chromatography (HPLC)-MS/MS measurements at the retention time of the reaction product between cholesterol-5α,6α-epoxide and N_α_-benzoylglycyl-l-lysine (*m/z* 710.5). (**A**) HPLC-MS/MS measurements of (a) cholesterol-5α,6α-epoxide, (b) N_α_-benzoylglycyl-l-lysine, and (c) a mixture of cholesterol-5α,6α-epoxide and N_α_-benzoylglycyl-l-lysine, each heated at 180 °C for 120 min; (**B**) product ion spectrum (parent ion, *m/z* = 710.5 ± 0.5; fragmentor voltage, 135 V; collision energy, 30 V) of the reaction product from the mixture of cholesterol-5α,6α-epoxide and N_α_-benzoylglycyl-l-lysine that had been heated at 180 °C for 120 min.

**Figure 6 foods-09-01882-f006:**
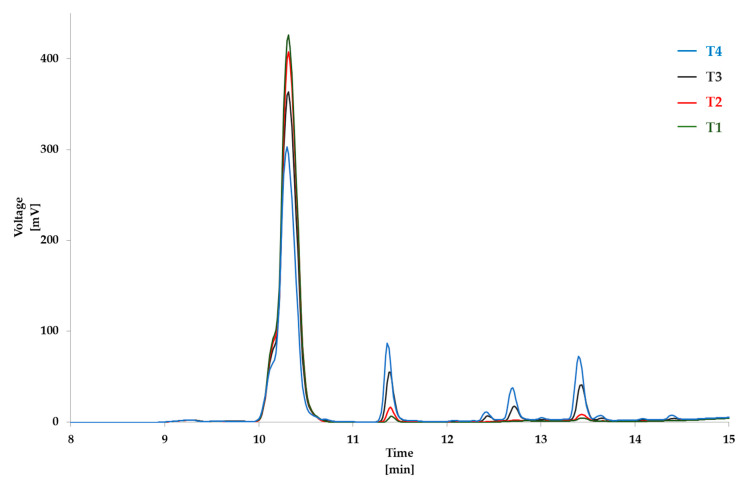
Overlaid chromatograms of N_α_-benzoylglycyl-l-lysine and derivatives in blank samples after different times of incubation at 180 °C (T1: 5 min, T2: 10 min, T3: 30 min, and T4: 60 min).

**Figure 7 foods-09-01882-f007:**
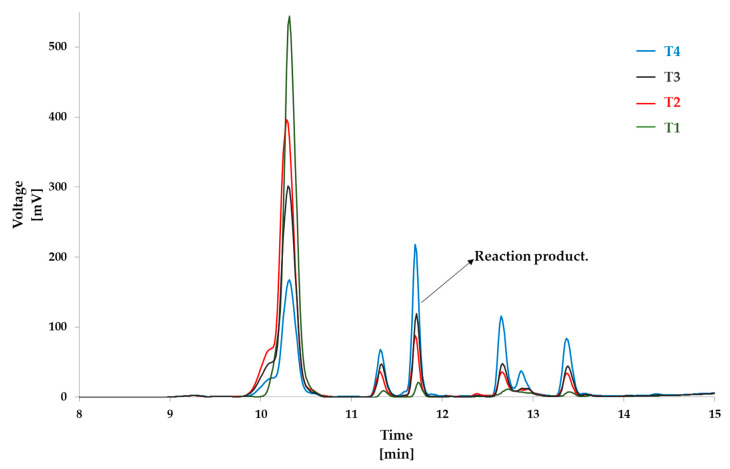
Overlaid chromatograms of N_α_-benzoylglycyl-l-lysine and derivatives in mixture reaction samples after different times of incubation at 180 °C (T1: 5 min, T2: 10 min, T3: 30 min, and T4: 60 min).

**Figure 8 foods-09-01882-f008:**
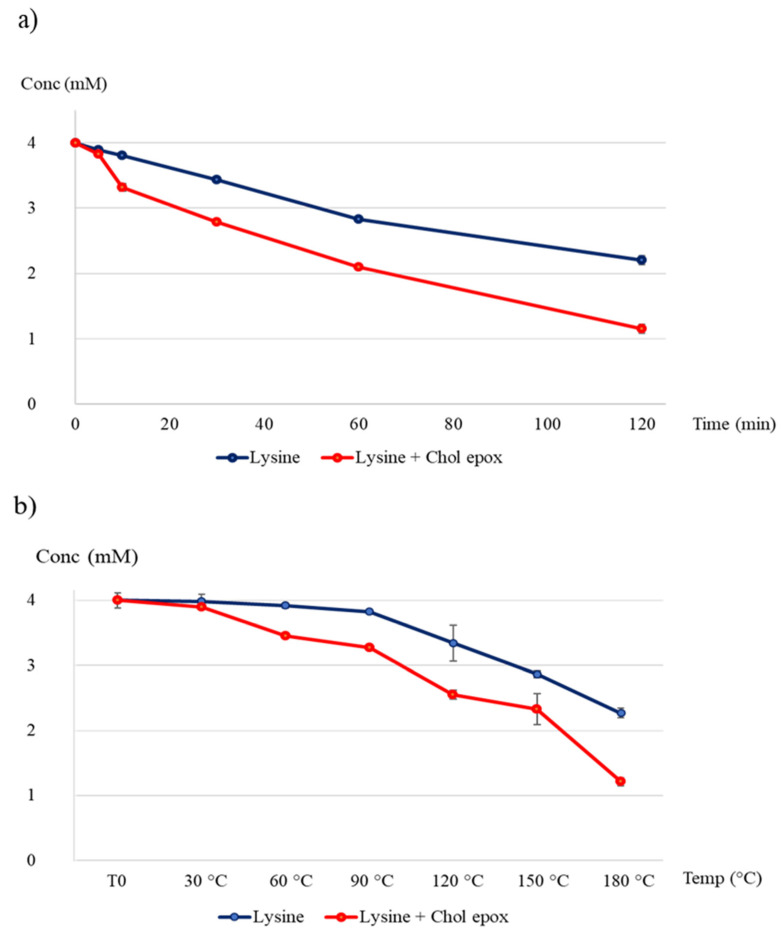
Kinetic studies: evolution of N_α_-benzoylglycyl-l-lysine levels in the reaction mixture (orange) and blank samples (blue). (**a**) Effect of time on the reaction between cholesterol 5α,6α-epoxide and the ε-amino-group of lysine (incubation temperature: 180 °C). (**b**) Effect of temperature on the reaction between cholesterol 5α,6α-epoxide and the ε-amino-group of lysine (incubation time: 120 min).

**Figure 9 foods-09-01882-f009:**
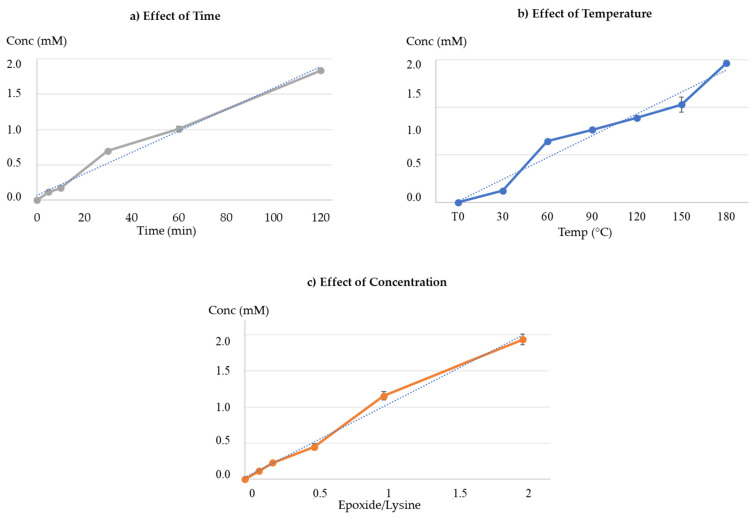
Kinetic studies: evolution of the reaction product levels from the reaction between N_α_-benzoylglycyl-l-lysine and cholesterol 5α,6α-epoxide. (**a**) Effect of time on the reaction between cholesterol 5α,6α-epoxide and the ε-amino-group of lysine (incubation temperature: 180 °C). (**b**) Effect of temperature on the reaction between cholesterol 5α,6α-epoxide and the ε-amino-group of lysine (incubation time:120 min). (**c**) Effect of oxysterol concentration (incubation at 180 °C for 120 min). Epoxide/Lysine = concentration of cholesterol 5α,6α-epoxide/concentration of N_α_-benzoylglycyl-l-lysine.

**Table 1 foods-09-01882-t001:** Method validation.

Compounds	Linear Range (mM)	Regression Line	R^2^	Reproducibility (RSD %, *n* = 5)		Sensitivity	
				Intra-Day	Inter-Day	LOD (µM)	LOQ (μM)
Cholesterol 5α,6α-epoxide	0.025–1.25	Y = 6.6399x − 0.1027	0.9992	0.6	0.8	4.5	15
N_α_-benzoylglycyl-l-lysine	0.05–1.0	Y = 8908.6x + 417.1	0.9953	1.9	3.9	0.3	1.1

R^2^: Coefficient of correlation, RSD: relative standard deviation, LOD: Limit of detection, and LOQ: Limit of quantification.

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
