# Peer review of "Food Protein Sterylation: Chemical Reactions between Reactive Amino Acids and Sterol Oxidation Products under Food Processing Conditions"

_foods, 2020, doi:10.3390/foods9121882_

Round 1
Reviewer 1 Report
Comments and remarks:
- the abstract part should be re-written, it is too chaotic, there is no details about highlights,
- the title is too general, please change it,
- the introduction part should be re-written,
- in the introduction part should be highlighted the main aim of the paper, and additionally, what is the novelty of carried research work,
- the list of abbreviation should be added,
- how do the Authors select the analytes? The rational of the choice of the selected biologically active compounds studied is missing and should be clearly discussed. Additionally, these analytes are not listed in the abstract section,
- what about validation of the obtained models (training and test sets)?,
- what is a value of mean square errors (MSE)?,
- it would be interesting to see data (e.g. chromatograms) from real biological samples but not spiked,
- for such experiments QTOF-MS/MS should be also used,
- the quality of figures is poor.
Author Response
Reviewer 1.
Comments and remarks:
- The abstract part should be re-written, it is too chaotic, there are no details about highlights,
Following the reviewer’s suggestion, the abstract has been re-written adding more details.
- The title is too general, please change it,
According to the reviewer’s suggestion, the title was improved.
- The introduction part should be re-written,
Following the reviewer’s suggestion, the introduction has been improved.
- In the introduction part should be highlighted the main aim of the paper, and additionally, what is the novelty of carried research work.
Following the reviewer’s comment, the main aim of the paper and the novelty of the research carried have been highlighted.
- The list of abbreviation should be added,
According to the reviewer’s comment, the list of abbreviation was added.
- How do the Authors select the analytes? The rationale of the choice of the selected biologically active compounds studied is missing and should be clearly discussed. Additionally, these analytes are not listed in the abstract section.
The rationale of the choice of the selected biologically active compounds was discussed in the article (4. Discussion). Analytes were added in the abstract section.
- What about validation of the obtained models (training and test sets)?,
The validation was obtained from the comparison of the results obtained from mixture models with blank samples. Moreover, analyses were performed in triplicate. To measure the significance of the difference between blank samples and mixture models was useless since no reaction product was observed in the blank sample.
- What is a value of mean square errors (MSE)?
This parameter was not measured.
- It would be interesting to see data (e.g. Chromatograms) from real biological samples but not spiked. For such experiments QTOF-MS/MS should be also used,
We agree with the suggestion of the reviewer. Furthers studies will be performed especially in food matrixes to identified phytosterylated proteins. These studies will require the use of more sophisticated mass analyzers such as QTOF-MS/MS. In this study, the formation of a compound from the reaction between cholesterol 5α,6α-epoxide and the free ε-amino-group of Nα-benzoylglycyl-l-lysine was demonstrated using ESI-MS/MS after TLC and HPLC-DAD observations. The presence of new compounds only observed in the mixture of cholesterol-5α,6α-epoxide and Nα-benzoylglycyl-l-lysine with m/z corresponding to the hypothesized reaction products evidenced that the sterylation reaction happened in the study conditions. In the next studies, a deepen characterization of other reaction products using QTOF will be performed.
- The quality of figures is poor.
The quality of figures has been improved.

Reviewer 2 Report
The manuscript sent by Nzekoue et al. entitled "Pilot studies on the chemical reaction in food between oxysterols/oxyphytosterols and proteins." concerns a very important and interesting topic in food chemistry. The article is written in a clear and understandable way and the authors have adopted a logical scientific approach. The selection of literature is appropriate and logical. Despite the fact that the work is written in a transparent way, the authors did not avoid several editorial errors.
Here are a few suggestions:
- All the figures are not very clear to readers. Please consider increasing the resolution.
- Figure 2, please provide some explanations on the peaks between 12 min and 13 min.
- Discussion, paragraph 4: it’s indented.
Author Response
Reviewer 2
The manuscript sent by Nzekoue et al. entitled "Pilot studies on the chemical reaction in food between oxysterols/oxyphytosterols and proteins." concerns a very important and interesting topic in food chemistry. The article is written in a clear and understandable way and the authors have adopted a logical scientific approach. The selection of literature is appropriate and logical. Despite the fact that the work is written in a transparent way, the authors did not avoid several editorial errors.
Here are a few suggestions:
1. All the figures are not very clear to readers. Please consider increasing the resolution.
Following the reviewer’s suggestion, the figures have been improved.
2. Figure 2, please provide some explanations on the peaks between 12 min and 13 min.
Compounds from 12-13 min could correspond to advanced sterylation products. Further studies will be performed for a deepen characterization of all the sterylation products and intermediates.
3. Discussion, paragraph 4: it’s indented.
